# Unsupervised Discovery of Temporal Structure in Noisy Data with Dynamical Components Analysis

**David G. Clark**[*,1,2]    **Jesse A. Livezey**[*,2,3]    **Kristofer E. Bouchard**[2,3,4]

dgc2138@cumc.columbia.edu    jlivezey@lbl.gov    kebouchard@lbl.gov

[*]Equal contribution.

[1]Center for Theoretical Neuroscience, Columbia University
[2]Biological Systems and Engineering Division, Lawrence Berkeley National Laboratory
[3]Redwood Center for Theoretical Neuroscience, University of California, Berkeley
[4]Helen Wills Neuroscience Institute, University of California, Berkeley

## Abstract

Linear dimensionality reduction methods are commonly used to extract low-dimensional structure from high-dimensional data. However, popular methods disregard temporal structure, rendering them prone to extracting noise rather than meaningful dynamics when applied to time series data. At the same time, many successful unsupervised learning methods for temporal, sequential and spatial data extract features which are predictive of their surrounding context. Combining these approaches, we introduce Dynamical Components Analysis (DCA), a linear dimensionality reduction method which discovers a subspace of high-dimensional time series data with maximal predictive information, defined as the mutual information between the past and future. We test DCA on synthetic examples and demonstrate its superior ability to extract dynamical structure compared to commonly used linear methods. We also apply DCA to several real-world datasets, showing that the dimensions extracted by DCA are more useful than those extracted by other methods for predicting future states and decoding auxiliary variables. Overall, DCA robustly extracts dynamical structure in noisy, high-dimensional data while retaining the computational efficiency and geometric interpretability of linear dimensionality reduction methods.

## 1    Introduction

Extracting meaningful structure from noisy, high-dimensional data in an unsupervised manner is a fundamental problem in many domains including neuroscience, physics, econometrics and climatology. In the case of time series data, e.g., the spiking activity of a network of neurons or the time-varying prices of many stocks, one often wishes to extract features which capture the dynamics underlying the system which generated the data. Such dynamics are often expected to be low-dimensional, reflecting the fact that the system has fewer effective degrees of freedom than observed variables. For instance, in neuroscience, recordings of 100s of neurons during simple stimuli or behaviors generally contain only ∼10 relevant dimensions [1]. In such cases, dimensionality reduction methods may be used to uncover the low-dimensional dynamical structure.

Linear dimensionality reduction methods are popular since they are computationally efficient, often reducing to generalized eigenvalue or simple optimization problems, and geometrically interpretable, since the high- and low-dimensional variables are related by a simple change of basis [2]. Analyzing the new basis can provide insight into the relationship between the high- and low-dimensional

---

DCA code is available at: https://github.com/BouchardLab/DynamicalComponentsAnalysis

variables [3]. However, many popular linear methods including Principal Components Analysis, Factor Analysis and Independent Components Analysis disregard temporal structure, treating data at different time steps as independent samples from a static distribution. Thus, these methods do not recover dynamical structure unless it happens to be associated with the static structure targeted by the chosen method.

On the other hand, several sophisticated unsupervised learning methods for temporal, sequential and spatial data have recently been proposed, many of them rooted in *prediction*. These prediction-based methods extract features which are predictive of the future (or surrounding sequential or spatial context) [4–9]. Predictive features form useful representations since they are generally linked to the dynamics, computation or other latent structure of the system which generated the data. Predictive features are also of interest to organisms, which must make internal estimates of the future of the world in order to guide behavior and compensate for latencies in sensory processing [10]. These ideas have been formalized mathematically [11, 12] and tested experimentally [13].

We introduce Dynamical Components Analysis (DCA), a novel method which combines the computational efficiency and ease of interpretation of linear dimensionality reduction methods with the temporal structure-discovery power of prediction-based methods. Specifically, DCA discovers a subspace of high-dimensional time series data with maximal predictive information, defined as the mutual information between the past and future [12]. To make the predictive information differentiable and accurately estimable, we employ a Gaussian approximation of the data, however we show that maximizing this approximation can yield near-optima of the full information-theoretic objective. We compare and contrast DCA with several existing methods, including Principal Components Analysis and Slow Feature Analysis, and demonstrate the superior ability of DCA to extract dynamical structure in synthetic data. We apply DCA to several real-world datasets including neural population activity, multi-city weather data and human kinematics. In all cases, we show that DCA outperforms commonly used linear dimensionality reduction methods at predicting future states and decoding auxiliary variables. Altogether, our results establish that DCA is an efficient and robust linear method for extracting dynamical structure embedded in noisy, high-dimensional time series.

## 2 Dynamical Components Analysis

### 2.1 Motivation

Dimensionality reduction methods that do not take time into account will miss dynamical structure that is not associated with the static structure targeted by the chosen method. We demonstrate this concretely in the context of Principal Components Analysis (PCA), whose static structure of interest is *variance* [14, 15]. Variance arises in time series due to both dynamics and noise, and the dimensions of greatest variance, found by PCA, contain contributions from both sources in general. Thus, PCA is prone to extracting spatially structured noise rather than dynamics if the noise variance dominates, or is comparable to, the dynamics variance (Fig. 1A). We note that for applications in which generic shared variability due to both dynamics and spatially structured noise is of interest, static methods are well-suited.

To further illustrate this failure mode of PCA, suppose we embed a low-dimensional dynamical system, e.g., a Lorenz attractor, in a higher-dimensional space via a random embedding (Fig. 1B,C). We then add spatially anisotropic Gaussian white noise (Fig. 1D). We define a signal-to-noise ratio (SNR) given by the ratio of the variances of the first principal components of the dynamics and noise. When the SNR is small, the noise variance dominates the dynamics variance and PCA primarily extracts noise, missing the dynamics. Only when the SNR becomes large does PCA extract dynamical structure (Fig. 1F,G, black). Rather than maximizing variance, DCA finds a projection which maximizes the mutual information between past and future windows of length $T$ (Fig. 1E). As we will show, this mutual information is maximized precisely when the projected time series contains as much dynamical structure, and as little noise, as possible. As a result, DCA extracts dynamical structure even for small SNR values, and consistently outperforms PCA in terms of dynamics reconstruction performance as the SNR grows (Fig 1F,G, red).

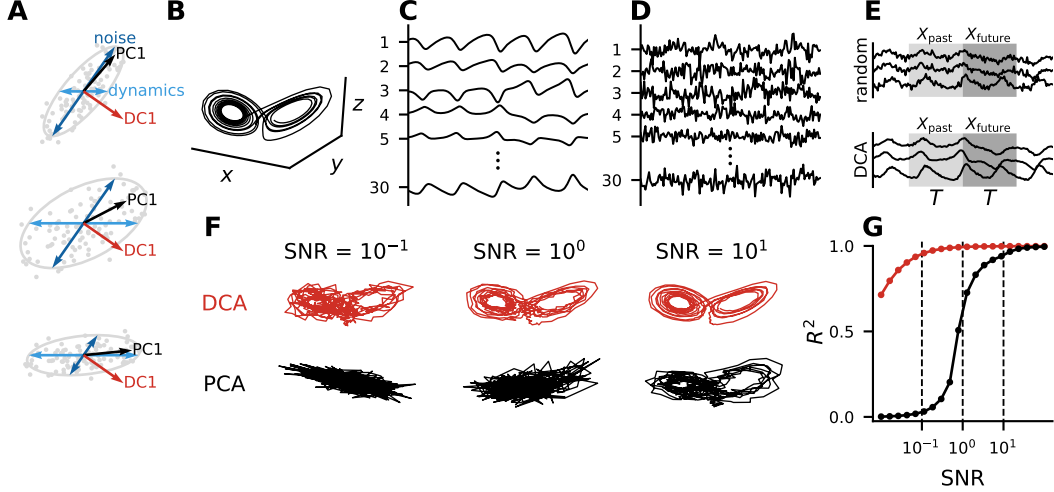

Figure 1: **DCA finds dynamics rather than variance. (A)** Schematic of unit vectors found by PCA and DCA for three relative levels of dynamics and noise. The dimension of greatest variance, found by PCA, contains contributions from both sources while the dimension found by DCA is orthogonal to the noise. **(B)** Lorenz attractor in the chaotic regime. **(C)** Random orthogonal embedding of the Lorenz attractor into 30-dimensional space. **(D)** Embedded Lorenz attractor with spatially-structured white noise. **(E)** Random three-dimensional projection (top) and DCA projection (bottom) of the embedded Lorenz attractor. **(F)** Reconstructions of the Lorenz attractor given the three-dimensional projections found by DCA and PCA. **(G)** Lorenz reconstruction performance ($R^2$) as a function of the SNR for both methods. See Appendix B for details of the noisy Lorenz embedding.

## 2.2   Predictive information as an objective function

The goal of DCA is to extract a subspace with maximal dynamical structure. One fundamental characteristic of dynamics is predictability: in a system with dynamics, future uncertainty is reduced by knowledge of the past. This reduction in future uncertainty may be quantified using information theory. In particular, if we equate uncertainty with entropy, this reduction in future uncertainty is the mutual information between the past and future. This quantity was termed predictive information by Bialek et al. [12]. Formally, consider a discrete time series $X = \{x_t\}$, $x_t \in \mathbb{R}^n$, with a stationary (time translation-invariant) probability distribution $P(X)$. Let $X_{\text{past}}$ and $X_{\text{future}}$ denote consecutive length-$T$ windows of $X$, i.e., $X_{\text{past}} = (x_{-T+1}, \ldots, x_0)$ and $X_{\text{future}} = (x_1, \ldots, x_T)$. Then, the predictive information $I_T^{\text{pred}}(X)$ is defined as

$$
\begin{aligned}
I_T^{\text{pred}}(X) &= H\left(X_{\text{future}}\right) - H\left(X_{\text{future}} | X_{\text{past}}\right) \\
&= H\left(X_{\text{past}}\right) + H\left(X_{\text{future}}\right) - H\left(X_{\text{past}}, X_{\text{future}}\right) \\
&= 2H_X(T) - H_X(2T)
\end{aligned}
\tag{1}
$$

where $H_X(T)$ is the entropy of any length-$T$ window of $X$, which is well-defined by virtue of the stationarity of $X$. Unlike entropy and related measures such as Kolmogorov complexity [16], predictive information is minimized, not maximized, by serially independent time series (white noise). This is because predictive information captures the sub-extensive component of the entropy of $X$. Specifically, if the data points that comprise $X$ are mutually independent, then $H_X(\alpha T) = \alpha H_X(T)$ for all $\alpha$ and $T$, meaning that the entropy is perfectly extensive. On the other hand, if $X$ has temporal structure, then $H_X(\alpha T) < \alpha H_X(T)$ and the entropy has a sub-extensive component given by $\alpha H_X(T) - H_X(\alpha T) > 0$. Upon setting $\alpha = 2$, this sub-extensive component is the predictive information.

Beyond simply being able to detect the presence of temporal structure in time series, predictive information discriminates between different types of structure. For example, consider two discrete-time Gaussian processes with autocovariance functions $f_1(\Delta t) = \exp\left(-|\Delta t/\tau|\right)$ and $f_2(\Delta t) = \exp\left(-\Delta t^2/\tau^2\right)$. For $\tau \gg 1$, the predictive information in these time series saturates as $T \to \infty$ to $c_1 \log \frac{\tau}{2}$ and $c_2 \tau^4$, respectively, where $c_1$ and $c_2$ are constants of order unity (see Ap-

pendix D for derivation). The disparity in the predictive information of these time series corresponds to differences in their underlying dynamics. In particular, $f_1(\Delta t)$ describes Markovian dynamics, leading to small predictive information, whereas $f_2(\Delta t)$ describes longer-timescale dependencies, leading to large predictive information. Finally, as discussed by Bialek et al. [12], the predictive information of many time series diverges with $T$. In these cases, different scaling behaviors of the predictive information correspond to different classes of time series. For one-dimensional time series, it was demonstrated that the divergent predictive information provides a unique complexity measure given simple requirements [12].

## 2.3 The DCA method

DCA takes as input samples $x_t \in \mathbb{R}^n$ of a discrete time series $X$, as well as a target dimensionality $d \leq n$, and outputs a projection matrix $V \in \mathbb{R}^{n \times d}$ such that the projected data $y_t = V^T x_t$ maximize an empirical estimate of $I_T^{\text{pred}}(Y)$. In certain cases of theoretical interest, $P(X)$ is known and $I_T^{\text{pred}}(Y)$ may be computed exactly for a given projection $V$. Systems for which this is possible include linear dynamical systems with Gaussian noise and Gaussian processes more broadly. In practice, however, we must estimate of $I_T^{\text{pred}}(Y)$ from finitely many samples. Directly estimating mutual information from multidimensional data with continuous support is possible, and popular nonparametric methods include those based on binning [17, 18], kernel density estimation [19] and $k$-nearest neighbor ($k$NN) statistics [20]. However, many of these nonparametric methods are not differentiable (e.g., $k$NN-based methods involve counting data points), complicating optimization. Moreover, these methods are typically sensitive to the choice of hyperparameters [21] and suffer from the curse of dimensionality, requiring prohibitively many samples for accurate results [22].

To circumvent these challenges, we assume that $X$ is a stationary (discrete-time) Gaussian process. It then follows that $Y$ is stationary and Gaussian since $Y$ is a linear projection of $X$. Under this assumption, $I_T^{\text{pred}}(Y)$ may be computed from the second-order statistics of $Y$, which may in turn be computed from the second-order statistics of $X$ given $V$. Crucially, this estimate of $I_T^{\text{pred}}(Y)$ is differentiable in $V$. Toward expressing $I_T^{\text{pred}}(Y)$ in terms of $V$, we define $\Sigma_T(X)$, the spatiotemporal covariance matrix of $X$ which encodes all second-order statistics of $X$ across $T$ time steps. Assuming that $\langle x_t \rangle_t = 0$, we have

$$\Sigma_T(X) = \begin{pmatrix} C_0 & C_1 & \dots & C_{T-1} \\ C_1^T & C_0 & \dots & C_{T-2} \\ \vdots & \vdots & \ddots & \vdots \\ C_{T-1}^T & C_{T-2}^T & \dots & C_0 \end{pmatrix} \quad \text{where} \quad C_{\Delta t} = \left\langle x_t x_{t+\Delta t}^T \right\rangle_t . \tag{2}$$

Then, the spatiotemporal covariance matrix of $Y$, $\Sigma_T(Y)$, is given by sending $C_{\Delta t} \to V^T C_{\Delta t} V$ in $\Sigma_T(X)$. Finally, $I_T^{\text{pred}}(Y)$ is given by

$$I_T^{\text{pred}}(Y) = 2H_Y(T) - H_Y(2T) = \log|\Sigma_T(Y)| - \frac{1}{2}\log|\Sigma_{2T}(Y)|. \tag{3}$$

To run DCA on data, we first compute the $2T$ cross-covariance matrices $C_0, \dots, C_{2T-1}$, then maximize the expression for $I_T^{\text{pred}}(Y)$ of Eq. 3 with respect to $V$ (see Appendix A for implementation details). Note that $I_T^{\text{pred}}(Y)$ is invariant under invertible linear transformations of the columns of $V$. Thus, DCA finds a subspace as opposed to an ordered sequence of one-dimensional projections.

Of course, real data violate the assumptions of both stationarity and Gaussianity. Note that stationarity is a fundamental conceptual assumption of our method in the sense that predictive information is defined only for stationary processes, for which the entropy as a function of window length is well-defined. Nonetheless, extensions of DCA which take nonstationarity into account are possible (see Discussion). On the other hand, the Gaussian assumption makes optimization tractable, but is not required in theory. Note, however, that the Gaussian assumption is acceptable so long as the optima of the Gaussian objective are also near-optima of the full information-theoretic objective. This is a much weaker condition than agreement between the Gaussian and full objectives over all possible $V$. To probe whether the weak condition might hold in practice, we compared the Gaussian estimate of predictive information to a direct estimate obtained using the nonparametric $k$NN estimator of Kraskov et al. [20] for projections of non-Gaussian synthetic data. We refer to

these two estimates of predictive information as the "Gaussian" and "full" estimates, respectively. For random one-dimensional projections of the three-dimensional Lorenz attractor, the Gaussian and full predictive information estimates are positively correlated, but show a complex, non-monotonic relationship (Fig. 2A,B). However, for one-dimensional projections of the 30-dimensional noisy Lorenz embedding of Fig. 1, we observe tight agreement between the two estimates for random projections (Fig. 2C, gray histogram). Running DCA, which by definition increases the Gaussian estimate of predictive information, also increases the full estimate (Fig. 2C, red trajectories). When we consider three-dimensional projections of the same system, random projections no longer efficiently sample the full range of predictive information, but running DCA nevertheless increases both the Gaussian and full estimates (Fig. 2D, trajectories). These results suggest that DCA finds good optima of the full, information-theoretic loss surface in this synthetic system despite only taking second-order statistics into account.

For a one-dimensional Gaussian time series $Y$, it is also possible to compute the predictive information using the Fourier transform of $Y$ [23]. In particular, when the asymptotic predictive information $I_{T \to \infty}^{\text{pred}}(Y)$ is finite, we have $I_{T \to \infty}^{\text{pred}}(Y) = \sum_{k=1}^{\infty} k b_k^2$ where $\{b_k\}$ are the so-called *cepstrum coefficients* of $Y$, which are related to the Fourier transform of $Y$ (see Appendix C). When the Fourier transform of $Y$ is estimated for length-$2T$ windows in conjunction with a window function, this method computes a regularized estimate of $I_T^{\text{pred}}(Y)$. We call this the "frequency-domain" method of computing Gaussian predictive information (in contrast the "time-domain" method of Eq. 3). Like the time-domain method, the frequency-domain method is differentiable in $V$. Its primary advantage lies in leveraging the fast Fourier transform (FFT), which allows DCA to be run with much larger $T$ than would be feasible using the time-domain method which requires computing the log-determinant of a $T$-by-$T$ matrix, an $\mathcal{O}(T^3)$ operation. By contrast, the FFT is $\mathcal{O}(T \log T)$. However, the frequency-domain method is limited to finding one-dimensional projections. To find a multidimensional projection, one can greedily find one-dimensional projections and iteratively project them out of of the problem, a technique called deflation. However, deflation is not guaranteed to find local optima of the DCA objective since correlations between the projected variables are ignored (Fig. 2E). For this reason, we use the time-domain implementation of DCA unless stated otherwise.

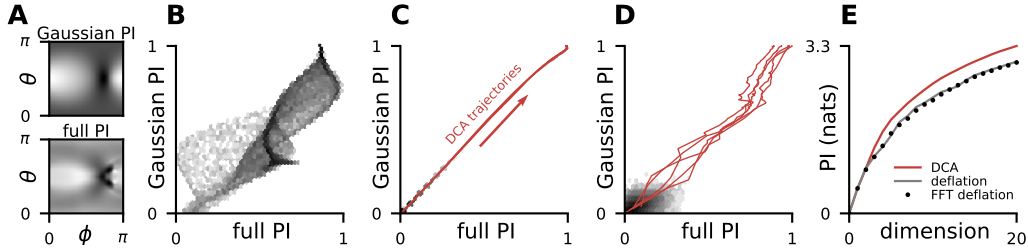

Figure 2: **Comparison of Gaussian vs. full predictive information estimates (A–D) and the frequency-domain method (E). (A)** Predictive information of one-dimensional projections of the three-dimensional Lorenz attractor as a function of the spherical coordinates $(\theta, \phi)$ of the projection using Gaussian and full ($k$NN) estimates. (A–D) all consider DCA with $T = 1$. **(B)** Histogram of the Gaussian and full estimates of predictive information from (A). **(C)** Histogram of the Gaussian and full estimates of predictive information of random one-dimensional projections of the 30-dimensional noisy Lorenz embedding of Fig. 1. Red trajectories correspond to five different runs of DCA. **(D)** Same as (C) but for three-dimensional projections of the same system. **(E)** Gaussian predictive information of subspaces found by different implementations of DCA when run on 109-dimensional motor cortical data (see Section 4). "DCA" directly optimizes Eq. 3, "deflation" optimizes Eq. 3 to find one-dimensional projections in a deflational fashion and "FFT deflation" uses the frequency-domain method of computing Gaussian predictive information in a deflational fashion. $T = 5$ is used in all three cases.

## 3 Related work

Though less common than static methods, linear dimensionality reduction methods which take time into account, like DCA, are sometimes used. One popular method is Slow Feature Analysis (SFA),

which we examine in some depth due to its resemblance to DCA [24, 25]. Given a discrete time sereis $X$, where $x_t \in \mathbb{R}^n$, SFA finds projected variables $y_t = V^T x_t \in \mathbb{R}^d$ that have unit variance, mutually uncorrelated components and minimal mean-squared time derivatives. For a discrete one-dimensional time series with unit variance, minimizing the mean-squared time derivative is equivalent to maximizing the one-time step autocorrelation. Thus, SFA may be formulated as

$$\text{maximize} \quad \text{tr}\left(V^T C_1^{\text{sym}} V\right) \text{ subject to } V^T C_0 V = I \qquad (4)$$

where $V \in \mathbb{R}^{n \times d}$, $C_0 = \langle x_t x_t^T \rangle_t$, $C_1 = \langle x_t x_{t+1}^T \rangle_t$ and $C_1^{\text{sym}} = \frac{1}{2}\left(C_1 + C_1^T\right)$. We assume that $X$ has been temporally oversampled so that the one-time step autocorrelation of any one-dimensional projection is positive, which is equivalent to assuming that $C_1^{\text{sym}}$ is positive-definite (see Appendix E for explanation). SFA is naturally compared to the $T = 1$ case of DCA. For one-dimensional projections ($d = 1$), the solutions of SFA and DCA coincide, since mutual information is monotonically related to correlation for Gaussian variables in the positive-correlation regime. For higher-dimensional projections ($d > 1$), the comparison becomes more subtle. SFA is solved by making the whitening transformation $\tilde{V} = C_0^{1/2} V$ and letting $\tilde{V}$ be the top-$d$ orthonormal eigenvectors of $M_{\text{SFA}} = C_0^{-1/2} C_1^{\text{sym}} C_0^{-1/2}$. To understand the solution to DCA, it is helpful to consider the relaxed problem of maximizing $I(U^T x_t; V^T x_{t+1})$ where $U$ need not equal $V$. The relaxed problem is solved by performing Canonical Correlation Analysis (CCA) on $x_t$ and $x_{t+1}$, which entails making the whitening transformations $\tilde{U} = C_0^{1/2} U$, $\tilde{V} = C_0^{1/2} V$ and letting $\tilde{U}$ and $\tilde{V}$ be the top-$d$ left and right singular vectors, respectively, of $M_{\text{CCA}} = C_0^{-1/2} C_1 C_0^{-1/2}$ [26, 27]. If $X$ has time-reversal symmetry, then $C_1^{\text{sym}} = C_1$, so $M_{\text{SFA}} = M_{\text{CCA}}$ and the projections found by SFA and DCA agree. For time-irreversible processes, $C_1^{\text{sym}} \neq C_1$, so $M_{\text{SFA}} \neq M_{\text{CCA}}$ and the projections found by SFA and DCA disagree. In particular, the SFA objective has no dependence on the off-diagonal elements of $V^T C_1 V$, while DCA takes these terms into account to maximize $I\left(V^T x_t; V^T x_{t+1}\right)$. Additionally, for non-Markovian processes, SFA and DCA yield different subspaces for $T > 1$ for all $d \geq 1$ since DCA captures longer-timescale dependencies than SFA (Fig. 3A). In summary, DCA is superior to SFA at capturing past-future mutual information for time-irreversible and/or non-Markovian processes. Note that most real-world systems including biological networks, stock markets and out-of-equilibrium physical systems are time-irreversible. Moreover, real-world systems are generally non-Markovian. Thus, when capturing past-future mutual information is of interest, DCA is superior to SFA for most realistic applications.

With regard to the relaxed problem solved by CCA, Tegmark [28] has suggested that, for time-irreversible processes $X$, the maximum of $I\left(U^T x_t; V^T x_{t+1}\right)$ can be significantly reduced when $U = V$ is enforced. This is because, in time-irreversible processes, predictive features are not necessarily predictable, and vice versa. However, because this work did not compare CCA (the optimal $U \neq V$ method) to DCA (the optimal $U = V$ method), the results are overly pessimistic. We repeated the analysis of [28] using both the noisy Lorenz embedding of Fig. 1 as well as a system of coupled oscillators that was used in [28]. For both systems, the single projection found by DCA captured almost as much past-future mutual information as the pair of projections found by CCA (Fig. 3B,C). This suggests that while predictive and predictable features are different in general, shared past and future features might suffice for capturing most of the past-future mutual information in a certain systems. Identifying and characterizing this class of systems could have important implications for prediction-based unsupervised learning techniques [28, 9].

In addition to SFA, other time-based linear dimensionality reduction methods have been proposed. Maximum Autocorrelation Factors [29] is equivalent to the version of SFA described here. Complexity Pursuit [30] and Forecastable Components Analysis [31] each minimize the entropy of a nonlinear function of the projected variables. They are similar in spirit to the frequency-domain implementation of DCA, but do not maximize past-future mutual information. Several algorithms inspired by Independent Components Analysis that incorporate time have been proposed [32–34], but are designed to separate independent dimensions in time series rather than discover a dynamical subspace with potentially correlated dimensions. Like DCA, Predictable Feature Analysis [35, 36] is a linear dimensionality reduction method with a prediction-based objective. However, Predictable Feature Analysis requires explicitly specifying a prediction model, whereas DCA does not assume a particular model. Moreover, Predictable Feature Analysis requires alternating optimization updates of the prediction model and the projection matrix, whereas DCA is end-to-end differentiable. Finally, DCA is related to the Past-Future Information Bottleneck [37] (see Appendix F).

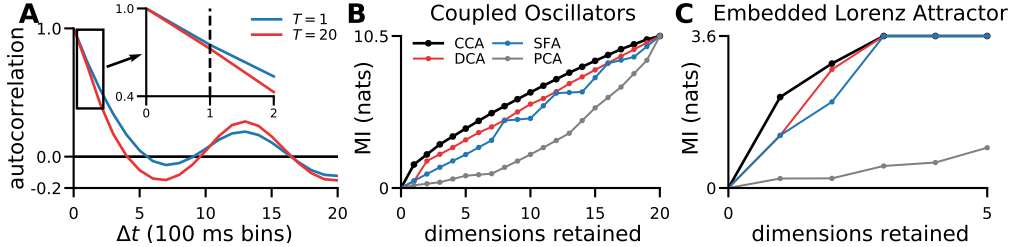

Figure 3: **Comparison of DCA with other methods.** (A) Autocorrelation functions of one-dimensional DCA projections of motor cortical data (see Section 4) for $T = 1$, in which case DCA is equivalent to SFA, and $T = 20$. While the one-time step autocorrelation is larger for the $T = 1$ projection (inset), the $T = 20$ projection exhibits stronger oscillations apparent at longer timescales. (B) Performance of DCA, SFA, PCA and CCA at capturing past-future mutual information, $I\left(U^T x_t; V^T x_{t+\Delta t}\right)$, where $U = V$ for DCA, SFA and PCA and $U \neq V$ for CCA. Following Tegmark [28], $x_t$ comprises the position and momentum variables of 10 coupled oscillators and $\Delta t = 10$. (C) Same as (B), but using the 30-dimensional noisy Lorenz embedding of Fig. 1 with $\Delta t = 2$.

We have been made aware of two existing methods which share the name Dynamical Component(s) Analysis [38–40]. Thematically, they share the goal of uncovering low-dimensional dynamics from time series data. Thirion and Faugeras [38] perform a two-stage, temporal then kernelized spatial analysis. Seifert et al. [39] and Korn et al. [40] assume the observed dynamics are formed by low-dimensional latent variables with linear and nonlinear dynamics. To fit a linear approximation of the latent variables, they derive a generalized eigenvalue problem which is sensitive to same-time and one-time step correlations, i.e., the data and the approximation of its first derivative.

An alternative to objective function-based components analysis methods are generative models, which postulate a low-dimensional latent state that has been embedded in high-dimensional observation space. Generative models featuring latent states imbued with dynamics, such as the Kalman filter, Gaussian Process Factor Analysis and LFADS, have found widespread use in neuroscience (see Appendix I for comparisons of DCA with the KF and GPFA) [41–43]. The power of these methods lies in the fact that rich dynamical structure can be encouraged in the latent state through careful choice of priors and model structure. However, learning and inference in generative models tend to be computationally expensive, particularly in models featuring dynamics. In the case of deep learning-based methods such as LFADS, there are often many model and optimization hyperparameters that need to be tuned. In terms of computational efficiency and simplicity, DCA occupies an attractive territory between linear methods like PCA and SFA, which are computationally efficient but extract relatively simple structure, and dynamical generative models like LFADS, which extract rich dynamical structure but are computationally demanding. As a components analysis method, DCA makes the desired properties of the learned features explicit through its objective function. Finally, the ability of DCA to yield a linear subspace in which dynamics unfold may be exploited for many analyses. For example, the loadings for DCA can be studied to examine the relationship between the high- and low-dimensional variables (Appendix J).

Lastly, while DCA does not produce an explicit description of the dynamics, this is a potentially attractive property. In particular, while dynamical generative models such as the KF provide descriptions of the dynamics, they also assume a particular form of dynamics, biasing the extracted components toward this form. By contrast, DCA is formulated in terms of spatiotemporal correlations and, as result, can extract broad forms of (stationary) dynamics, be they linear or nonlinear. For example, the Lorenz attractor of Fig. 1 is a nonlinear dynamical system.

## 4   Applications to real data

We used DCA to extract dynamical subspaces in four high-dimensional time series datasets: (i) multi-neuronal spiking activity of 109 single units recorded in monkey primary motor cortex (M1) while the monkey performed a continuous grid-based reaching task [44]; (ii) multi-neuronal spiking activity of 55 single units recorded in rat hippocampal CA1 while the rat performed a reward-chasing

task [45, 46]; (iii) multi-city temperature data from 30 cities over several years [47]; and (iv) 12 variables from an accelerometer, gyroscope, and gravity sensor recording human kinematics [48]. See Appendix B for details. For all results, three bins of projected data were used to predict one bin of response data. Data were split into five folds, and reported $R^2$ values are averaged across folds.

To assess the performance of DCA, we noted that subspaces which capture dynamics should be more predictive of future states than those which capture static structure. Moreover, for the motor cortical and hippocampal datasets, subspaces which capture dynamics should be more predictive of behavioral variables (cursor kinematics and rat location, respectively) than subspaces which do not, since neural dynamics are believed to underlie or encode these variables [49, 50]. Thus, we compared the abilities of subspaces found by DCA, PCA and SFA to decode behavioral variables for the motor cortical and hippocampal datasets and to forecast future full-dimensional states for the temperature and accelerometer datasets.

For the motor cortical and hippocampal datasets, DCA outperformed PCA at predicting both current and future behavioral variables on held-out data (Fig. 4, top row). This reflects the existence of dimensions which have substantial variance, but which do not capture as much dynamical structure as other, smaller-variance dimensions. Unlike PCA, DCA is not drawn to these noisy, high-variance dimensions. In addition to demonstrating that DCA captures more dynamical structure than PCA, this analysis demonstrates the utility of DCA in a common task in neuroscience, namely, extracting low-dimensional representations of neural dynamics for visualization or further analysis (see Appendix H for forecasting results on the neural data and Appendix J for example latent trajectories and their relationship to the original measurement variables) [27, 51]. For the temperature dataset, DCA and PCA performed similarly, and for the accelerometer dataset, DCA outperformed PCA for the lowest-dimensional projections. The narrower performance gap between DCA and PCA on the temperature and accelerometer datasets suggests that the alignment between variance and dynamics is stronger in these datasets than in the neural data.

Assuming Gaussianity, DCA is formally superior to SFA at capturing past-future mutual information in time series which are time-irreversible and/or non-Markovian (Section 3). All four of our datasets possess both of these properties, suggesting that subspaces extracted by DCA might offer superior decoding and forecasting performance to those extracted by SFA. We found this to be the case across all four datasets (Fig. 4, bottom row). Moreover, the relative performance of DCA often became stronger as $T$ (the past-future window size of DCA) was increased, highlighting the non-Markovian nature of the data (see Appendix G for absolute $R^2$ values). This underscores the importance of leveraging spatiotemporal statistics across long timescales when extracting non-Markovian dynamical structure from data.

## 5   Discussion

DCA retains the geometric interpretability of linear dimensionality reduction methods while implementing an information-theoretic objective function that robustly extracts dynamical structure while minimizing noise. Indeed, the subspace found by DCA may be thought of as the result of a competition between aligning the subspace with dynamics and making the subspace orthogonal to noise, as in Fig. 1A. Applied to neural, weather and accelerometer datasets, DCA often outperforms PCA, indicating that noise variance often dominates or is comparable to dynamics variance in these datasets. Moreover, DCA often outperforms SFA, particularly when DCA integrates spatiotemporal statistics over long timescales, highlighting the non-Markovian statistical dependencies present in these datasets. Overall, our results show that DCA is well-suited for finding dynamical subspaces in time series with structural attributes characteristic of real-world data.

Many extensions of DCA are possible. Since real-world data generation processes are generally non-stationary, extending DCA for non-stationary data is a key direction for future work. For example, non-stationary data may be segmented into windows such that the data are approximately stationary within each window [52]. In general, the subspace found by DCA includes contributions from all of the original variables. For increased interpretability, DCA could be optimized with an $\ell^1$ penalty on the projection matrix $V$ [53] to identify a small set of relevant features, e.g., individual neurons or stocks [3]. Both the time- and frequency-domain implementations of DCA may be made differentiable in the input data, opening the door to extensions of DCA that learn nonlinear transformations of the input data, including kernel-like dimensionality expansion, or that use a nonlinear mapping from the

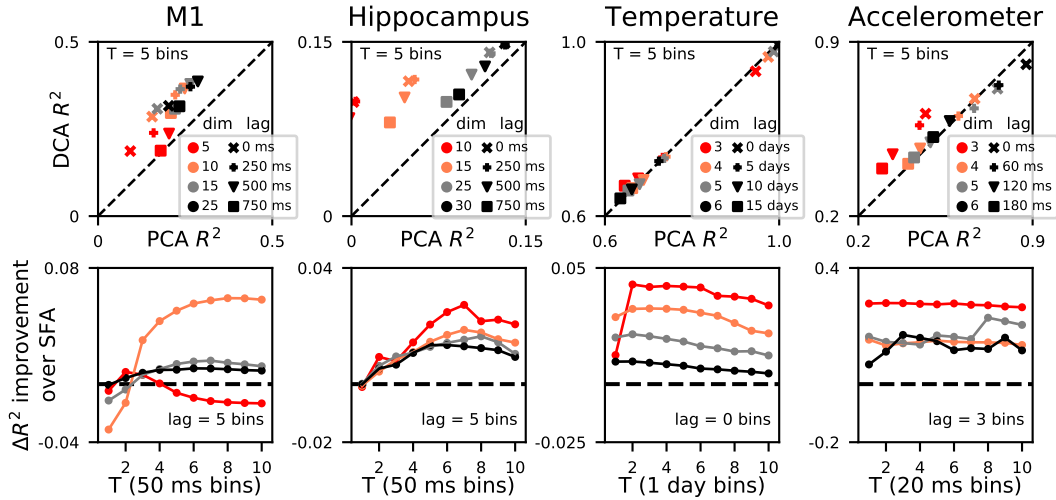

Figure 4: **DCA for prediction and forecasting.** For all panels, color indicates the projected dimensionality. For the top row, marker type indicates the lag for prediction. The top row compares held-out $R^2$ for DCA vs. PCA as a function of projection dimensionality and prediction lag. The bottom row shows the difference in held-out $R^2$ for DCA vs. SFA as a function of $T$, the past-future window size parameter for DCA. (**M1**) Predicting cursor location from projected motor cortical data. (**Hippocampus**) Predicting animal location from projected hippocampal data. (**Temperature**) Forecasting future full-dimensional temperature states from projected temperature states. (**Accelerometer**) Forecasting future full-dimensional accelerometer states from projected states.

high- to low-dimensional space, including deep architectures. Since DCA finds a linear projection, it can also be kernelized using the kernel trick. The DCA objective could also be used in recurrent neural networks to encourage rich dynamics. Finally, dimensionality reduction via DCA could serve as a preprocessing step for time series analysis methods which scale unfavorably in the dimensionality of the input data, allowing such techniques to be scaled to high-dimensional data.

## Acknowledgements

D.G.C. and K.E.B. were funded by LBNL Laboratory Directed Research and Development. We thank the Neural Systems and Data Science Lab and Laurenz Wiskott for helpful discussion.

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
