[Supplementary Material]

# Appendix for: Unsupervised Discovery of Temporal Structure in Noisy Data with Dynamical Components Analysis

**David G. Clark, Jesse A. Livezey, Kristofer E. Bouchard**

## A  Implementation, optimization & time complexity

DCA was implemented in PyTorch and optimized using the L-BFGS-B algorithm in SciPy [1–3]. All analysis was run on a desktop computer. To make optimization more stable, we included a penalty term in the loss function which encourages the projection matrix $V$ to have orthonormal columns during optimization. The final loss is given by

$$\mathcal{L}_{\mathrm{DCA}} = -I_T^{\mathrm{pred}}(Y) + \lambda \left\| V^T V - I \right\|_F^2 \qquad (1)$$

where $\lambda > 0$. The orthonormality penalty is always exactly zero at the end of optimization since mutual information is invariant under invertible transformations of either of its arguments, so we can always transform $V$ to have orthonormal columns without changing the predictive information. Because the loss function is non-convex, we typically perform 5 random initializations.

For large spatial or temporal dimensions with large correlations, the spatiotemporal covariance matrix was occasionally not positive-definite. In these cases, we added a constant to the diagonal of $C_0$, the same-time covariance matrix, so that the smallest eigenvalue was 1e-6 (equivalent to adding a small amount of uncorrelated noise to all dimensions for all times).

To fit DCA for a dataset with total length $T_{\mathrm{tot}}$, dimensionality $n$ and projection dimensionality $d$ using past and future windows of length $T$, the first step is to compute the spatiotemporal covariance matrix, which is $\mathcal{O}\left(n^2 T^2 T_{\mathrm{tot}}\right)$. The time complexity of DCA's optimization procedure does not scale in the total amount of data since the objective references the data only through this spatiotemporal covariance matrix which is computed prior to optimization. Each evaluation of the objective, or its gradient, requires computing $2T$ quadratic products of the form $V^T C_{\Delta t} V$, each of which is $\mathcal{O}\left(n^2 d + nd^2\right)$, as well as the log-determinants, which are $\mathcal{O}\left(T^3 d^3\right)$. Altogether, each evaluation is $\mathcal{O}\left(Tn^2 d + Tnd^2 + T^3 d^3\right)$. For all datasets we considered, optimization time dominated the time required to compute the spatiotemporal covariance matrix.

## B  Datasets and preprocessing

### B.1  Lorenz attractor synthetic data

The governing equations of the Lorenz attractor are [4]

$$\begin{aligned} \dot{x} &= \sigma\,(y - x) \\ \dot{y} &= x\,(\rho - z) - y \\ \dot{z} &= xy - \beta z. \end{aligned} \qquad (2)$$

In all appearances of the Lorenz attractor in the main text, we used the parameters $\sigma = 10$, $\beta = \frac{8}{3}$ and $\rho = 28$, which place the system in the chaotic regime. We used an integration time step of $\Delta t = 5 \times 10^{-3}$, then downsampled the data by a factor of 5.

For the 30-dimensional noisy embedded Lorenz attractor used in Figures 1-3, we first generated three-dimensional Lorenz data as described above, then embedded the dynamics into 30-dimensional space via a random orthogonal embedding. We then added Gaussian white noise, which we parameterized by the eigenspectrum and eigenvectors of its covariance matrix. In particular, the eigenspectrum was given by

$$\lambda_j = \sigma^2 \exp\left(-\frac{2j}{d_{\mathrm{noise}}}\right) \qquad (3)$$

where $\sigma^2$ controls the overall amount of noise and $d_{\mathrm{noise}}$ controls the effective dimensionality of the noise. In all cases, we used $d_{\mathrm{noise}} = 7$. In Fig. 1, the parameter $\sigma^2$, which is also the variance of the first principal component of of the noise, was varied to obtain different SNR values. In Figures 2 and 3, $\sigma^2$ was fixed to achieve an SNR of unity. The eigenvectors of the noise covariance were chosen uniformly at random with the constraint the leading $d_{\mathrm{noise}}$ eigenvectors had close-to-median principal angles with respect to the subspace containing the Lorenz attractor.

### B.2 Monkey motor cortical dataset

O'Doherty et al. [5] released multi-electrode spiking data for both M1 and S1 for two monkeys during a continuous grid-based reaching task. We used M1 data from the subject "Indy" (specifically, we used the file "indy_20160627_01.mat"). We discarded single units with fewer than 5,000 spikes, leaving 109 units. We binned the spikes into non-overlapping bins (100 ms in Fig. 3, 50 ms in Figures 2 and 4), square-root transformed the data and mean-centered the data using a sliding window 30 s in width.

### B.3 Rat hippocampal data

Glaser et al. [6] released the original data of Mizuseki et al. [7] (dataset "hc2", session "ec014.333"). The data consist of 93 minutes of extracellular recordings from layer CA1 of dorsal hippocampus while a rat chased rewards on a square platform. We discarded single units with fewer than 10 spikes, leaving 55 units. We binned the spikes into non-overlapping 50 ms bins, then square-root transformed the data.

### B.4 Temperature dataset

The temperature dataset consists of hourly temperature data for 30 U.S. cities over a period of 7 years from `OpenWeatherMap.org` [8]. We downsampled the data by a factor of 24 to obtain daily tempeartures.

### B.5 Accelerometer dataset

Malekzadeh et al. [9] released accelerometer data which records roll, pitch, yaw, gravity $\{x, y, z\}$, rotation $\{x, y, z\}$ and acceleration $\{x, y, z\}$ for a total of 12 kinematic variables. The sampling rate is 50 Hz. We used the file "sub_19.csv" from "A_DeviceMotion_data.zip".

## C  Frequency-domain predictive information computation

Consider a one-dimensional discrete-time Gaussian process $Y$ with zero mean and autocovariance function $f(\Delta t)$. We can use Theorem 2.5 from Li and Xie [10] (see also [11]) to compute the predictive information of $Y$ in terms of the discrete-time Fourier transform (DTFT) of $f(\Delta t)$. Specifically, this theorem says that when the asymptotic predictive information $I_{T \to \infty}^{\mathrm{pred}}(Y)$ is finite, we have

$$I_{T \to \infty}^{\mathrm{pred}}(Y) = \sum_{k=1}^{\infty} k b_k^2, \tag{4}$$

as stated in the main text. The numbers $\{b_k\}$ are called the cepstrum coefficients of $Y$. They comprise the DTFT of the logarithm of the DTFT of $f(\Delta t)$:

$$b_k = \frac{1}{2\pi} \int_{-\pi}^{\pi} d\lambda e^{-i\lambda k} \log \tilde{f}(\lambda), \quad \tilde{f}(\lambda) = \sum_{k=-\infty}^{\infty} e^{-i\lambda k} f(k). \tag{5}$$

In practice, rather than directly computing the autocovariance function $f(\Delta t)$ of $Y$ and taking its DTFT, we compute the power spectral density of $Y$, which is equivalent to the DTFT of $f(\Delta t)$. Specifically, we use the FFT in conjunction with a window function to compute the power spectral density in many length-$2T$ windows of $Y$, then average the results together. If $f(\Delta t)$ falls off to zero with a timescale $\tau \ll 2T$, then this method computes the full asymptotic predictive information $I_{T \to \infty}^{\mathrm{pred}}(Y)$ in the limit of infinite samples. If $f(\Delta t)$ does not fall off this quickly, then the window function effectively forces the autocovariance to decay to zero at $\Delta t = \pm 2T$, yielding a regularized estimate of $I_T^{\mathrm{pred}}(Y)$.

# D Asymtotic predictive information derivations

## D.1 Exponential autocovariance

Let $Y_1$ be a discrete-time Gaussian process whose autocovariance function $f_1(\Delta t)$ is an exponential:

$$f_1(\Delta t) = \exp\left(-\left|\frac{\Delta t}{\tau}\right|\right). \tag{6}$$

It is easy to show that $f_1(\Delta t)$ is the autocovariance function of an AR(1) process

$$y_t = Ay_{t-1} + e_t \tag{7}$$

where $\langle e_t^2 \rangle_t = \Omega^2$ and $\langle e_t e_{t+\Delta t} \rangle_t = 0$ for $|\Delta t| \geq 1$. The autocovariance of this process is

$$\mathbf{E}[y_t y_{t+\Delta t}] = \frac{\Omega^2}{1 - A^2} A^{-|\Delta t|}, \tag{8}$$

and if we set $\Omega^2 = 1 - e^{\frac{2}{\tau}}$ and $A = e^{\frac{1}{\tau}}$ we have

$$\mathbf{E}[y_t y_{t+\Delta t}] = e^{-\frac{|k|}{\tau}} = f_1(\Delta t). \tag{9}$$

Since this process is Markovian, $I_T^{\text{pred}}(Y_1)$ is simply the mutual information between two consecutive time steps for all $T \geq 1$. Thus, we have

$$I_{T\to\infty}^{\text{pred}}(Y_1) = -\frac{1}{2}\log\left(1 - f_1(1)^2\right) = -\frac{1}{2}\log\left(1 - e^{-\frac{2}{\tau}}\right) \tag{10}$$

and for $\tau \gg 1$, this becomes

$$I_{T\to\infty}^{\text{pred}}(Y_1) = \frac{1}{2}\log\frac{\tau}{2}. \tag{11}$$

## D.2 Squared-exponential autocovariance

Let $Y_2$ be a discrete-time Gaussian process whose autocovariance function $f_2(\Delta t)$ is a squared-exponential:

$$f_2(\Delta t) = \exp\left(-\frac{\Delta t^2}{\tau^2}\right). \tag{12}$$

We will compute the predictive information using the cepstrum coefficients. The DTFT $\tilde{f}_2(\lambda)$ of $f_2(\Delta t)$ is

$$\tilde{f}_2(\lambda) = \sum_{k=-\infty}^{\infty} \cos(k\lambda) f_2(k) = \sqrt{\pi}\tau e^{-\frac{1}{4}\lambda^2\tau^2} \sum_{k=-\infty}^{\infty} e^{-\tau^2\left(k^2\pi^2 + k\pi\lambda\right)}. \tag{13}$$

Taking the log gives

$$\log\tilde{f}_2(\lambda) = \frac{1}{2}\sqrt{\pi} + \log\tau - \frac{1}{4}\lambda^2\tau^2 + \log\left(1 + e^{-\tau^2(\pi^2 + \pi\lambda)} + e^{-\tau^2(\pi^2 - \pi\lambda)} + \cdots\right). \tag{14}$$

For $\tau \gg 1$, the logarithmic term on the RHS is approximately $\log 1 = 0$ (this is not true close to the endpoints $\lambda = \pm\pi$ where this term is approximately $\log 2$, however $\log\tilde{f}_2(\lambda)$ also contains a term quadratic in $\tau$ which dominates the logarithmic term close to $\lambda = \pm\pi$). Taking another DTFT gives the cepstrum coefficients:

$$b_k \approx \frac{1}{\pi}\int_0^\pi \cos(k\lambda)\left(\frac{1}{2}\sqrt{\pi} + \log\tau - \frac{1}{4}\lambda^2\tau^2\right)d\lambda = -\frac{\tau^2\cos(k\pi)}{2k^2}. \tag{15}$$

Thus, we have

$$I_{T\to\infty}^{\text{pred}}(Y_2) = \frac{1}{2}\sum_{k=1}^{\infty} kb_k^2 = \frac{\tau^4}{8}\sum_{k=1}^{\infty}\frac{1}{k^3} = \frac{\zeta(3)}{8}\tau^4 \approx 0.15026 \times \tau^4 \tag{16}$$

where $\zeta$ is the Riemann zeta function.

## E  Note on Slow Feature Analysis

If we allow for one-dimensional projections of $X$ with negative one-time step autocorrelation $\rho_1 < 0$, then the one time-step mutual information $I_1$ is non-monotonically related to $\rho_1$ across different projections according to $I_1 = -\frac{1}{2} \log\left(1 - \rho_1^2\right)$. As a result, SFA is no longer guaranteed to coincide with DCA for $d = 1$, nor for time-reversible processes with $d \geq 1$. However, if we modify SFA to order the projections according to decreasing $\rho_1^2$, rather than decreasing $\rho_1$, then these guarantees are restored. The notion that slowness ($\rho_1 > 0$) and fastness ($\rho_1 < 0$) are equivalent in the eyes of mutual information was pointed out by Creutzig and Sprekeler [12], who presented the information-theoretic interpretation of SFA for time-reversible processes.

## F  Note on the Past-Future Information Bottleneck

The problem of finding representations of time series that optimally capture past-future mutual information has been studied in the context of the Information Bottleneck (IB), a method for compressing data in a way that retains its relevant aspects [13]. In particular, the Past-Future Information Bottleneck (PFIP) seeks a compressed representation $Y$ of $X_{\text{past}}$ that has maximal mutual information with $X_{\text{future}}$, subject to a fixed amount of compression [12, 14, 15]. This corresponds to a variational problem with the Lagrangian

$$\mathcal{L}_{\text{PFIB}} = I\left(X_{\text{past}}; Y\right) - \beta I\left(Y; X_{\text{future}}\right), \tag{17}$$

where $\beta$ controls the tradeoff between compression of $X_{\text{past}}$ and prediction of $X_{\text{future}}$. The most fundamental difference between the PFIB and DCA is that the PFIB compresses only the past, whereas DCA compresses both the past and the future by projecting to a lower-dimensional space. However, there is nonetheless a case in which the PFIB and DCA solutions coincide.

When the "observed" and "relevant" variables in an IB problem ($X_{\text{past}}$ and $X_{\text{future}}$ in the PFIB) are jointly Gaussian, then the solution is closely related to CCA [16]. As $\beta$ is increased, the solution undergoes a cascade of structural phase transitions which increase the dimensionality of the compressed representation, i.e., the number of CCA components retained. For one-time step past and future windows, the PFIB solution is to retain the top-$d$ left singular vectors of $C_0^{-1/2} C_1 C_0^{-1/2}$, where $d$ is determined by $\beta$. For time-reversible processes, the PFIB solution coincides with that of DCA (and with SFA [12]). This is potentially surprising, since the PFIB maximizes $I\left(y_t; x_{t+1}\right)$ while DCA maximizes $I\left(y_t; y_{t+1}\right)$. Put differently, in Gaussian processes with time-reversal symmetry, the features of the past which are the most self-predictive are also the most predictive overall. For time-irreversible processes, DCA, SFA and the PFIB admit mutually distinct solutions: we show in the main text that DCA and SFA disagree, and the PFIB solution must be different from those of both DCA and SFA since the solutions to both of these methods are invariant under time-reversal transformations while the solution to the PFIB is not.

## G  Absolute $\mathrm{R}^2$ values

Fig 1 shows the absolute held-out $R^2$ values for the DCA vs. SFA comparison in Fig 4 of the main text. Note that SFA does not depend on $T$.

## H  Neural Forecasting

For the M1 and hippocampus datasets in Fig 4 of the main text, we also ran the forecasting analysis in which the projected neural state is used to predict future full-dimensional neural states. Fig 2 shows comparisons of DCA with PCA and SFA. At short time lags, PCA is expected to have higher $R^2$ due to the optimality of PCA at capturing the Frobenius norm. At longer time lags, DCA, PCA, and SFA all have relatively low predictive power.

## I  Kalman Filter and Gaussian Process Factor Analysis

Two popular dynamical generative models used to infer latent dynamics from time series data are the Kalman Filter (KF) and Gaussian Process Factor Analysis (GPFA). The KF assumes that a latent

Figure 1: **Absolute $R^2$.** Absolute held-out $R^2$ values for the DCA vs. SFA comparisons in Fig 4 of the main text. See main text for legends.

Figure 2: **Neural forecasting.** For the M1 and hippocampus datasets, the held-out $R^2$ is shown across dimension and lags.

time series with linear, Gaussian dynamics has been linearly embedded into observation space with Gaussian observation noise. Similarly, GPFA assumes that a latent time series whose components are independent Gaussian processes with a common kernel, but independent timescales, has been linearly embedded into observation space with Gaussian observation noise. In both cases, the model parameters are fit using the expectation-maximization (EM) algorithm [17, 18]. To infer the latent state at time $t$, each model admits both a causal procedure, which uses observations at times $t' \in [1, \dots, t]$, as well as a non-causal procedure, which uses observations at times $t' \in [1, \dots, T]$. For the KF, the causal and non-causal procedures are called Kalman filtering and smoothing, respectively. For each of the four datasets analyzed in the main text, we inferred latent states using the KF and GPFA using both causal and non-causal inference procedures using 5-fold cross validation. Note that while both the causal and non-causal procedures for each method use many observations to infer each time step of the latent dynamics, DCA uses only one. Thus, performance comparisons between DCA and the KF or GPFA are somewhat ill-posed since the latent factors for DCA incorporate less information. However, between the causal and non-causal inference procedures, the causal procedures

are better suited for comparison to DCA since their resulting latent factors do not incorporate future observations.

The KF model was fit using its EM algorithm, as derived in Ghahramani and Hinton [17]. During the E step, which computes the parameters of the Gaussian distribution over the latent states using forward and backward passes, we employed a steady-state optimization in which we did not recompute matrices which had converged to their steady-state values during each pass[1]. Latent factors inferred using both the causal and non-causal inference procedures for the KF performed better than factors extracted using DCA at decoding behavioral variables from neural data, and the performance gap was largest for large numbers of factors and short time lags (Fig. 3, M1 and Hippocampus). For the non-neural datasets, factors extracted using DCA generally had better forecasting performance, particularly for the accelerometer dataset (Fig. 3, Temperature and Accelerometer).

Figure 3: **DCA vs. the Kalman filter.** For all panels, color indicates the number of factors and marker type indicates the lag for prediction (see Fig. 4 of the main text for legends). Each panel compares held-out $R^2$ for DCA vs. PCA as a function of the number of factors and prediction lag. Top row: causal inference procedure (Kalman filtering). Bottom row: non-causal inference procedure (Kalman smoothing).

We used the MATLAB code accompanying Yu et al. [18] for EM and inference in the GPFA model. For all datasets, the data were segmented into non-overlapping "trials" of 100 time steps, which provided substantial speedups. For performance evaluation, factors extracted by DCA were segmented to have the same trial structure as those inferred using GPFA and care was taken to not evaluate decoding or forecasting performance across trial boundaries. EM was initialized using factor analysis and run for 300 iterations, which we confirmed was adequate for convergence based on inspection of the log-likelihood over training. The provided implementation of GPFA automatically segmented our trials of length 100 into shorter segments of length 20 during fitting. Latent factors inferred using the causal procedure for GPFA performed slightly better than factors extracted using DCA at decoding behavioral variables from neural data (Fig. 4, M1 and Hippocampus, top row) while the performance gap for the non-causal inference procedure was larger (Fig. 4, M1 and Hippocampus, bottom row). For the non-neural datasets, factors extracted using DCA performed slightly better than those inferred using either the causal or non-causal procedures for GPFA (Fig. 4, Temperature and Accelerometer).

## J  Inferred states and leverage scores

For each of the four datasets considered, we visualized the extracted latent states after three-dimensional projections found using DCA and PCA. Fig 5 shows the projections from DCA (top

Figure 4: **DCA vs. GPFA.** For all panels, color indicates the number of factors and marker type indicates the lag for prediction (see Fig. 4 of the main text for legends). Each panel compares held-out $R^2$ for DCA vs. PCA as a function of the number of factors and prediction lag. Top row: causal inference procedure. Bottom row: non-causal inference procedure.

panels in pairs) compared to the projections from PCA (bottom panel in pairs). Since DCA yields a subspace rather than an ordered sequence of components, we transformed the DCA projections using PCA so that the DCA components were ordered by variance explained, making the comparison to PCA more clear. The Spearman's rank-correlation between the top 3 components of DCA and PCA are: M1 (0.98, 0.76, 0.3), HC (0.42, 0.15, 0.12), Temperature (1.0, 0.93, 0.78) and Accelerometer (0.89, 0.88, -0.11). As expected, the DCA projections are typically lower amplitude compared to the PCA projections. This is exaggerated in the hippocampus dataset. In the accelerometer dataset, the DCA projections are smoother than their PCA counterparts.

The DCA and PCA subspaces can also be compared through their leverage scores [19] which measure the level of alignment between a subspace and the original measurement axes. Given an orthonormal basis $V \in \mathbb{R}^{n \times d}$ for a $d$-dimensional subspace, the leverage score for measurement axis $j$ is

$$\pi_j = \frac{1}{d} \sum_{i=1}^{d} (v_{ij})^2 \tag{18}$$

where, from the definition, $\sum_j \pi_j = 1$ and $0 \le \pi_j \le 1$. This means that $\pi_j$ can be thought of as a distribution.

Fig 6 shows the sorted leverage scores and a comparison of the DCA and PCA leverage scores across measurement axes. For all datasets except for the hippocampus dataset, the DCA leverage scores have a sharper peak compared to the PCA leverage scores. For the neural datasets, the DCA leverage scores have a larger tail compared to the PCA leverage scores. On the temperature and accelerometer datasets, DCA has a few measurement axes with large leverage scores and a slowly decaying tail while the PCA leverage scores decay approximately linearly. Only on the M1 dataset are the DCA and PCA leverage scores significantly correlated as measured by the Spearman's rank-correlation (RC).

Figure 5: **Inferred latent states for DCA vs. PCA.** For each dataset, the projections into a three-dimensional space are shown for DCA and PCA. The projections have been ordered by variance explained in the subspace but are plotted with unit-normalized variance.

Figure 6: **Leverage scores for the DCA and PCA subspaces.** The left panels show the distribution of leverage scores when they are sorted individually for DCA and PCA. The right panels show the DCA leverage scores vs. the PCA leverage scores per measurement axis (log-log scale). The Spearman's rank-correlation (RC) and significance is inset for each dataset.

## Footnotes

[1]Our Kalman filter EM code is available at: `https://github.com/davidclark1/FastKF`