[Reviews · NeurIPS 2019]

Reviewer 1



I'm pretty enthusiastic about this paper, though I can imagine shortcomings of this method -- particularly for non-stationary time series, and latent dynamics with nonlinear mappings. Despite these potential limitations, the method appears highly useful to me, given the prevalence of dimensionality reduction methods in neural data analysis. Further, this paper opens the door to future work relaxing assumptions of stationarity and linearity, and does an excellent job of summarizing the current state of the field and related statistical methods.

Reviewer 2



What are the black dashed lines in Fig 3? How are the subspaces discovered by DCA particularly interpretable? What do they "look" like, for instance in any of the real-world datasets show in Fig. 4? Would visualizing any of them be illuminating about underlying structures and mechanisms beyond what would have been possible with, say, PCA? How does the method scale with dimensionality and duration of the data? The frequency-domain method outlined is very cool.

Reviewer 3



I appreciated the author's responses, and I think the proposed refinements will strengthen the manuscript. As such, I decided to increase my score: 5 -> 6. However, I remain lukewarm regarding the actual results shown in the paper. I found the comparisons to be limited (the authors still resist performance comparisons with common approaches such as GPFA or LDS in Fig. 4), and the performance quantifications to not be very elucidating (they are focused solely on modest gains in predictive performance whereas the strongest motivation for this method is interpretability). Given that what I find most exciting in this submission is the potential for interpretability, I'm pretty disappointed no effort is done to explore this avenue in the results. To be clear, I agree that it is unreasonable to expect fully featured scientific results in a NeurIPS submission, but I would have liked to at least see this interpretability aspect briefly explored. =============================================================== The authors propose a new linear dimensionality reduction method, termed Dynamical Components Analysis (DCA), which finds low-dimensional projections of observed data by taking into account its temporal structure, namely the dependence between successive time points in the inferred latent space. The authors then validate the proposed method by applying it to two sets of neuronal data as well as two other real world datasets, focusing on the ability of the extracted projections to explain external variables and future states. Quality and Clarity The authors tackle an interesting problem, and motivate their approach well. I do think the description of PCA’s shortcomings is a bit extreme in the Abstract and Introduction sections. Indeed, in Neuroscience applications, practitioners are often interested in shared variability across neurons, something PCA and Factor Analysis are adept at describing. There is of course value in capturing temporal structure, but I find it a stretch to suggest the method proposed here is strictly superior to PCA or FA. Rather, they have different objectives. Furthermore, the method proposed is designed to capture temporal variability, but fails to provide any proper dynamical description, i.e., given activity at time t, this method makes no predictions regarding activity in the future (other than seeking to maximize dependency across time), neither does it offer any approximation to the dynamical rules at play. This is in contrast to, for example, a kalman filter/linear dynamical systems approach, which provides a linear approximation to the underlying dynamics. I think it would be interesting to discuss this limitation in the paper. From a technical standpoint, the authors did a great job at describing DCA in section 2. Section 2.1 offers a fair comparison with PCA, and it is very motivating to see DCA do so well at capturing temporal structure in this example. Sections 2.2 and 2.3 were clear and explored the technical choices well. The comparisons with SFA and CCA in section 3 are interesting, but I feel this section could benefit from taking a higher level, more intuitive approach. As the authors state, one could try to capture dependencies across time using CCA between consecutive time points. This has the advantage that it does not assume that the predictive and predictable subspaces are the same (V != U), which DCA does. This is indeed an important point, as V != U for large families of dynamical systems, and in fact CCA outperforms DCA at capturing mutual information across time in this section. At this point, a reader will be wondering why one should use DCA over CCA. The fact that CCA returns two subspaces (predictive and predicted) doesn’t seem like a big downfall. It is my interpretation that DCA takes into account dependencies over the entire sequence (across all time), which this CCA application does not. It would be interesting to see this play out in practice. In particular, it was unclear to me why CCA was not used in section 4. Also, while the authors mention GPFA and the Kalman filter as alternative approaches, these methods are absent in the comparisons in section 4. I think it would have been extremely helpful to include them. The authors state: “learning and inference in generative models tend to be computationally expensive, particularly in models featuring dynamics, and there are often many model and optimization hyperparameters that need to be tuned.” This is certainly true for LFADS, but it’s a bit of a stretch when referring to GPFA and the Kalman filter. It DCA is that much faster and easier to fit, it would be good to show this by direct comparison. Finally, it was not clear to me why in Fig. 4 performance for the neuronal datasets was measured only with respect to external variables. I agree that it is an interesting quantification, but it should come in addition to the simpler and more direct predictability of future states. Furthermore, the results shown in Fig. 4 are a bit underwhelming: the relative improvement over SFA is pretty small across datasets, and DCA seem to struggle to outperform PCA in most cases. Originality The model proposed here is, to the best of my knowledge, novel in this context. Significance While I think the method proposed here is interesting and has the potential to be useful in practice, the experimental part of the manuscript needs to be improved for this work to have a significant impact. As it stands, it is unclear how this method compared to popular alternatives, such as GPFA, Kalman filter or temporal CCA, and it is not obvious it presents a significant improvement over PCA in the real world datasets. Furthermore, while the authors tout the ability of DCA to extract interpretable projections, no attempt is made to explore or interpret these.

[Author Response · NeurIPS 2019]

We thank the reviewers for their feedback. We will first respond to shared and then to individual comments.

All of the reviewers expressed interest in further comparisons between DCA and other methods, the subject of Section 4. We agree and will include several additional results: 1) comparisons with GPFA and the KF in the analyses of Fig. 4, 2) forecasting results for the neural datasets, 3) visualizations of the extracted DCA and PCA components and 4) a comparison of the loadings, i.e., the magnitudes of the components of the projection matrices, found by DCA and PCA.

Additionally, reviewers 2 and 3 requested clarification regarding the advantages of DCA over other methods. The key advantages of DCA over generative models such as the KF, GPFA and LFADS stem from the fact that DCA is a linear components-analysis method, i.e., it uses a time-independent linear projection for dimensionality reduction. The following specific advantages will be more clearly articulated in the revised paper. First, the loadings for DCA can be inspected to interpret the relationship between the high- and low-dimensional variables, as in the analysis we will include. For instance, one could attempt to correlate each neuron's contribution to the DCA subspace with single-neuron properties, such as its predictive information. Second, it is easy to augment DCA with an $\ell_1$ penalty to achieve sparse feature selection, whereas sparse selection is much harder to incorporate in generative models. This could be used to select a small number of "important" neurons in population data, e.g., for BMI control. Third, one often wants to interpret the extracted components as being computed by a linear readout neuron. Finally, DCA can be kernelized to achieve nonlinear dimensionality reduction. Studying the behavior of Kernel DCA is a direction for future studies.

Additionally, we found and corrected a minor bug in Fig. 3A: the SFA and DCA lines are now blue and red, respectively.

**Reviewer 1:** While the GPFA model features a time-independent linear mapping from the latent space into the observation space, mapping each observation into the latent space (i.e., inference) uses the observations across all of time. Thus, dimensionality reduction using GPFA does not take the form of a time-independent linear projection as in DCA, PCA and SFA, and for this reason we do not lump GPFA in with these methods.

**Reviewer 2:** The dashed lines in Fig. 3 were intended to help visualization, but will be removed. The time complexity of DCA's fitting procedure does not scale in the total amount of data since DCA only needs spatiotemporal correlations which are computed prior to optimization. Each evaluation of the objective, or its gradient, requires computing $\mathcal{O}\left(T\right)$ quadratic products of the form $V^T C_{\Delta t} V$, each of which is $\mathcal{O}\left(n^2 d + n d^2\right)$, as well as the log determinants, which are $\mathcal{O}\left(T^3\right)$. Altogether, each evaluation is $\mathcal{O}\left(T n^2 d + T n d^2 + T^3\right)$. We will include this analysis.

**Reviewer 3:** We agree that DCA is not inherently superior to static methods, e.g., PCA and FA, but rather extracts a particular type of structure. Indeed, in neuroscience applications in which generic shared variability due to both dynamics and spatially structured noise is of interest, static methods are well-suited. We will clarify this.

It is true that DCA does not produce an explicit description of the dynamics. However, this is a potentially attractive property: while dynamical generative models such as the KF provide descriptions of the dynamics, they also assume a particular form of dynamics, biasing the extracted components toward this form. By contrast, DCA is formulated in terms of spatiotemporal correlations and, as result, can extract broad forms of (stationary) dynamics, be they linear or nonlinear. For example, the Lorenz attractor is a nonlinear dynamical system. That said, an interesting avenue for future work involves using DCA as a preprocessing step for methods which produce interpretable descriptions of the dynamics. For example, Burton et al. proposed a method for extracting nonlinear ODEs governing multidimensional time series, however this method scales poorly in the time series dimensionality (PNAS, 2016). To circumvent this challenge, DCA could be used to define a small number of dynamical degrees of freedom which are simple linear combinations of the original variables. Alternatively, the $\ell_1$-regularized version of DCA could be used to select a small number of original variables which capture the majority of the temporal structure.

CCA differs from DCA in two key ways: 1) CCA finds distinct past and future subspaces and 2) CCA does not capture long-timescale dynamics. With regard to the first difference, note that in most applications, one is interested in finding a single subspace. Moreover, the two subspaces provided by CCA are only interpretable as a pair: the features in $U$ are predictive of the features in $V$ and vice versa. This is why we did not include CCA in our comparisons. With regard to the second difference, note that interesting dynamics are sometimes only discernible at long timescales (Fig. 3A).

We agree that GPFA and the KF do not have excessively many hyperparameters and will clarify this. With regard to computational efficiency, note that DCA does not scale in the total amount of data, whereas GPFA, the KF and LFADS do. Direct comparisons of fitting time are challenging since each method has a tolerance parameter, and it is unclear how to configure these parameters for a fair comparison.

For the DCA vs. PCA comparisons, the fact that the former outperforms the latter to a greater extent on some datasets and to a lesser extent on others reflects the underlying structure of the datasets, and this is potentially interesting. For the DCA vs. SFA comparisons, although the change in $R^2$ values between the two methods looks small, these values should be compared to the raw $R^2$ value for each method, which themselves are quite small due to the difficulty of the task and the fact that we are doing simple linear regression. The raw $R^2$ values will be included in the final manuscript.

[Meta-Review · NeurIPS 2019]

While a couple of the reviews were short and hence not fully justifying their eventual scorings, the paper does have novel ideas and looks at an important problem with experiments to back it up. And the reviwers are also in agreement that the paper deserves to be accepted.